# Fertilizer profitability for smallholder maize farmers in Tanzania: A spatially-explicit *ex ante* analysis

**Sebastian Palmas** *, **Jordan Chamberlin**

International Maize and Wheat Improvement Center (CIMMYT), Nairobi, Kenya

* palmasforest@gmail.com

**Data Availability Statement:** The code used for the analysis presented in this paper is available for download and improvement from GitHub (github.com/spalmas/Ex-Ante).

## Abstract

We present an easily calibrated spatial modeling framework for estimating location-specific fertilizer responses, using smallholder maize farming in Tanzania as a case study. By incorporating spatially varying input and output prices, we predict the expected profitability for a location-specific smallholder farmer. A stochastic rainfall component of the model allows us to quantify the uncertainty around expected economic returns. The resulting mapped estimates of expected profitability and uncertainty are good predictors of actual smallholder fertilizer usage in nationally representative household survey data. The integration of agronomic and economic information in our framework makes it a powerful tool for spatially explicit targeting of agricultural technologies and complementary investments, as well as estimating returns to investments at multiple scales.

## Introduction

Smallholder farming systems of sub-Saharan Africa are characterized by persistently low productivity levels. While there has been some growth in recent years, most of this has come from area expansion rather than yield gains, and average yield gaps remain about 80% [1]. Increasing mineral fertilizer application is generally accepted as a fundamental component of strategies to address this productivity gap [2]. But levels of fertilizer usage, and application levels by those who do use fertilizer, generally remain low across the region [3].

Why are fertilizer usage levels so low? A rich empirical literature has developed in recent years, which emphasizes three key constraints:—agronomic responses to fertilizer are often much lower in farmers' fields than on researcher-managed trials, and such responses are substantially variable over geographic space [4–8]. Low and variable agronomic returns translate into low and variable economic returns once considering the local farm-gate crop and fertilizer prices. A number of empirical studies document such fertilizer profitability patterns for maize farmers in SSA [9–15]. (See [16] for a recent review of over 20 studies estimating the profitability of applying inorganic fertilizer on maize in various African locations.) Fourth, the stochastic nature of agricultural production in the absence of insurance markets means that small farmers face high variability of expected returns [17]. Given risk averseness, such uncertain

**Funding:** Support for this study was provided by the Bill & Melinda Gates Foundation, through the Taking Maize Agronomy to Scale in Africa (TAMASA) project (grant no: OPP1113374); from a grant from the U.S. Agency for International Development (USAID) via the Geospatial and Farming Systems Consortium led by the University of California at Davis; and from the CGIAR Research Program MAIZE, led by the International Maize and Wheat Improvement Centre (CIMMYT).

**Competing interests:** The authors have declared that no competing interests exist.

returns may represent powerful disincentives to invest even where the expected returns are relatively high [17–19].

But while profitability and risk are longstanding components of agricultural economists' evaluations of this question, there has been a dearth of planning and targeting frameworks for fertilizer that integrate biophysical responses with information about profitability, crop management and riskiness of returns(although there have been a number of very important investments in spatially explicit soils information—e.g. the Tanzanian Soil Information System TanSIS—which could inform such integrative frameworks). This is problematic because without such frameworks governments and private sector actors may struggle to identify optimal areas to focus market development activities—i.e., areas where farmers will likely face the most substantial gains to fertilizer investments—or to coordinate complementary investments—e.g., promoting fertilizer alongside crop insurance or other risk-reducing financial instruments. Furthermore, the fact that locally optimal fertilizer recommendations for any particular crop may vary considerably across locations argues for planning frameworks that allow for spatial variation in agronomic responses [20].

The goal of this paper is to present a spatially explicit framework for evaluating the likely economic and agronomic returns to fertilizer investments by smallholder farmers (and the uncertainty around those returns). We use data from smallholder maize farmers in Tanzania to parameterize an empirical model, and then implement that model within a spatially explicit environment that takes account of the spatial distributions of soil and rainfall characteristics, farmers and farmland, and input and output prices. Tanzania is a useful case study because it is emblematic in many ways of the broader adoption issues in the region. Recent nationally representative data indicate that only 15% of smallholder farmers use fertilizer, at an average rate of <70 kg/ha (authors' calculations from the 2008/09, 2010/11 and 2012/13 waves of the Tanzanian LSMS-ISA surveys). Considerable efforts have been made by the Tanzanian government to stimulate demand and facilitate access to fertilizers, including the National Agricultural Input Voucher Scheme (NAIVS), which provided fertilizer at subsidized rates between 2008/9 and 2013/14. One of the goals of NAIVS was to facilitate a relatively low-risk learning opportunity around fertilizer for farmers, which was expected to translate to eventual increases in market demand [21]. Understanding and planning for such demand will depend in part on better tools for estimating the economic returns to fertilizer at market prices.

We show that there is substantial variation in local yield responses and that after incorporating local price ratios for maize and nitrogen fertilizer, even larger variability in economic returns over space. We show that such spatial variability in returns is a useful predictor of actual farmer fertilizer usage. Furthermore, the role of stochastic rainfall is similarly highly variable across the country, varying in ways that differ from the distribution of expected profitability. Furthermore, we show that the returns to locally-optimized fertilizer recommendations (as opposed to national-level blanket recommendations) appear to be substantial and may represent important ways of raising aggregate economic returns to fertilizer investments at the farming system level.

We provide the data and code necessary to replicate our results and to implement similar frameworks in other settings (i.e., other countries, crops or inputs). We argue that greater usage of such approaches to evaluating the potential economic returns to fertilizer—as well as other production technologies promoted by international R&D institutions—will help to address the disconnect between agricultural technology R4D and farmer decision-making on the ground.

## Spatial ex ante analysis framework

### Overview

This section outlines a spatial framework that integrates biophysical and socio-economic variables measured over broad spatial scales (Fig 1). A key idea is that if we are able to reasonably predict yield responses to fertilizer as a function of spatially varying predictors, then we have a basis for building a spatially explicit evaluation framework. In principle, such a response function could be based on a structural model, such as QUEFTS [22] or could take various parametric or non-parametric approaches to empirical prediction. The only requirements are that (a) the agronomic response predictions are reasonably good, and (b) we have a sufficient set of geospatial model covariates to serve as out of sample predictors within similar geographies.

In the current era, we have increasing amounts of georeferenced agronomic response data to work with, even in traditionally data-sparse environments and a similarly broad set of modeling approaches. In this analysis, we take an empirical approach, defining a Random Forest model on a georeferenced dataset of small farmer maize yields and associated agronomic management data, as well as soils, terrain, rainfall and other biophysical parameters taken from geospatial datasets in the public domain. We describe these in more detail in the next section. While our modeling focus is on yield responses to nitrogen, in principle, any other agronomic response that has a coherent spatial expression could be modeled in this way, including any agronomic responses conditioned by soils, terrain, rainfall, temperature, etc.

A second key idea is the incorporation of spatially varying input and output prices. Smallholder farmers in SSA operate within large and heterogeneous market access contexts, with farm-gate prices varying considerably from location to location (e.g., [23–25]). Recognition of this is important to any efforts to meaningfully evaluate technology attractiveness from the farmer's perspective. Despite the absence of local market price data, we show that modeling approaches for predicting local prices in spatially coherent ways are feasible.

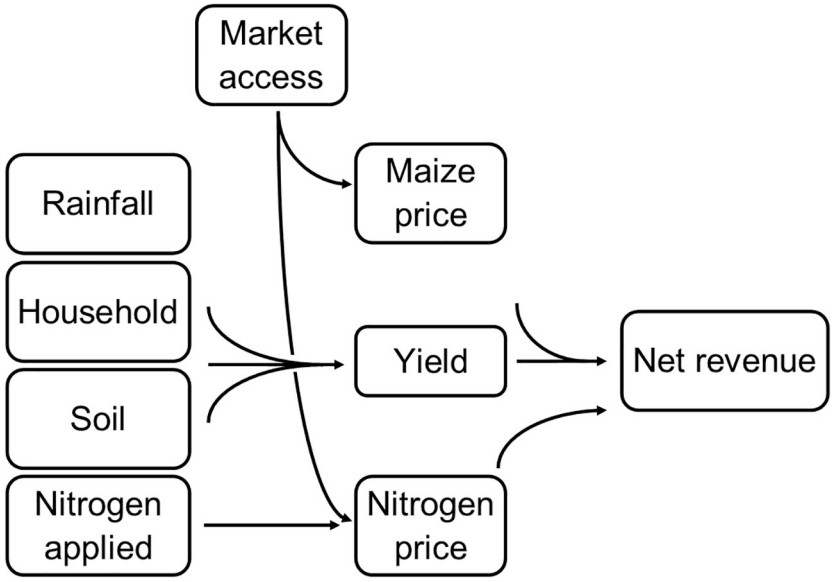

**Fig 1. Framework overview.**

A third key conceptual feature of our approach is based on accounting for the stochasticity of responses. Smallholder farmers are famously risk-averse, and abundant empirical evidence suggests that risk is an important element of smallholder decision making [26–28]. In our implementation, we achieve this through the inclusion of seasonal rainfall parameters. However, other sources of spatially varying uncertainty could also be incorporated, such as price volatility, which is known to vary with remoteness [29]. Model output can then be defined as a function of the stochastic parameters.

Linking all these elements, we have a framework for evaluating (a) an expected site-specific agronomic response—in this case: maize yield responses to nitrogen fertilizer, (b) the expected profitability of such input use, under local input and output prices and other assumptions, and (c) the uncertainty around these expected returns at any given location. Given the availability of databases on the distribution of rural populations, cropland and production, we can aggregate up model output to evaluate the likely aggregate benefits. Such guidance is critical for policymakers and development partners who must allocate scarce resources to meeting strategic rural development goals.

To carry out the analysis in this paper, we used R 4.0.2 [30]. Random forest models were constructed using the "randomForest" R package. Least-cost distances to estimate market access were calculated using the "gdistance" package.

## Modeling yield response to fertilizer

To model maize yield responses to nitrogen, we used the Tanzania Agronomy Panel Survey (APS) on 553 households in 25 districts of Tanzania collected during the main maize harvest periods of 2016 and 2017 [31]. Because of lack of measurements in the field, only 601 yield estimates from 455 households were available for modeling. The districts in our study were selected based on representativeness of favorable maize production defined as (1) areas with suitable research and extension partners that would allow the scaling of fertilizer decision support tools, (2) areas with extensive coverage of maize producing areas as classified by the Africa Soil Information Service—AfSIS, and (3) areas with relatively high human population densities (i.e., $>25/km^2$) with good access to urban markets (within 4 hrs of travel time). Georeferenced maize yields were measured using crop cuts (see [32]), and are accompanied by detailed data on plot characteristics, agronomic management (including fertilizer applications), and other household, farm and farm manager characteristics. The crop cut protocol involved collecting a grain sample, which was dried to 15% moisture before weighing. An aggregate N application rate for the plot was calculated on the basis of all the fertilizer applications reported by the farmer for that field—i.e., recorded across multiple fertilizer types and application rates, and normalized by the size of the field.

To estimate maize yield responses to nitrogen fertilizer, we employed a Random Forest model, a machine learning approach for diagnostics and prediction [33, 34]. In addition to nitrogen and crop management household variables from the survey data, spatial estimates of elevation, slope, soil organic carbon and pH, and total seasonal rainfall were included as predictors. Altitude and slope were obtained from the CGIAR-SRTM 90m digital elevation model Version 4.1 [35]. Soil property maps for organic carbon and pH at a 250m resolution soil came from the soil prediction surfaces from the AfSIS project [36]. We calculated seasonal (December to May) rainfall for each household from monthly predictions at 4 $km^2$ resolution from the TAMSAT v3.0 database [37].

The averages of the household variables found in the survey data were used to simulate the yields countrywide (Table 1). The model was validated using bootstrap sampling and partial dependency plots were reviewed for theoretical coherence.

**Table 1. Mean and standard deviation (SD) of the Tanzania Agronomy Panel Survey (APS) data.**

| Variable | Mean | Standard Deviation |
|---|---|---|
| Maize yield (kg/ha) | 2604.0 | 1832.8 |
| Fertilizer use (yes = 1) | 0.357 | 0.479 |
| N application rate among fertilizer users (kg/ha) | 35.2 | 98.0 |
| P application rate among fertilizer users (kg/ha) | 11.5 | 45.4 |
| Intercrop (yes = 1) | 0.573 | 0.4950 |
| Crop rotation (yes = 1) | 0.062 | 0.2407 |
| Use of manure (yes = 1) | 0.203 | 0.4028 |
| Use of crop residue (yes = 1) | 0.090 | 0.2864 |
| Number of weedings | 1.827 | 0.5542 |
| Use of improved seeds (yes = 1) | 0.148 | 0.3557 |
| Field in fallow in the last 3 years (yes = 1) | 0.040 | 0.1961 |
| Erosion control structure (yes = 1) | 0.245 | 0.4304 |
| Terraced field (yes = 1) | 0.035 | 0.1839 |
| Area in hectares of focal plot (log) | -0.507 | 0.9556 |
| Age of head of household | 47.702 | 13.7150 |
| Household size (Number of persons) | 5.692 | 3.1144 |
| Years of education of head of household | 7.067 | 3.5461 |
| Households | 455 | |
| Observations | 601 | |

Values pooled across years. The table shows the average and standard deviation of values in the farm survey data, which were used to estimate yield responses.

## Spatial prices estimation

Local farm-gate prices for maize in unsampled locations were estimated with a model that captures information on geographic location, as well as pixel-level meteorological and other environmental conditions, and market access characteristics, an approach similar to that described by [38]. Wholesale market prices for maize were obtained from the 4[th] wave of the Tanzania LSMS National Panel Survey 2014–2015 [39]. Original prices in TZS/kg were transformed to USD/kg using an exchange rate TZS 1598 to the US Dollar. We used this exchange rate throughout the analysis. The 601 spatially located observed prices were used to fit a random forest model using predictor variables capturing aspects of market access (travel time to market and distance to port), potential demand (population density and cropland) and precipitation averages, as well as longitude and latitude. Table A in S1 Text describes the complete list of variables used in the maize market price model.

From the estimated local market prices, we predict farm-gate maize prices by assuming a "last mile" transport cost rate of 0.01 USD/kg/hr. Specifically, a farm-gate price is estimated for every grid location as the highest of all possible farm-gate prices obtainable from different market locations, after accounting for the market-specific transportation costs between the market and the farm location (i.e., the embedded assumption is that a farmer will sell her production to the market which gives the highest price, after accounting for the costs required to transport output to that market).

For nitrogen application, we start with a representative market price of 0.95 USD per kg of Ng, which we derive from the average price of urea (generally the cheapest source of N) reported from AfricaFertilizer.org for Tanzania over the last five years. The price for nitrogen was inferred from the urea price, on the basis of the 46% N content of urea. We estimated

farm-gate nitrogen prices by fitting a logistic transportation cost model under which delivered fertilizer prices treble at 5 hours of travel time from a local market (Fig A in S1 Text).

A key predictor of local input and output prices was the estimated travel time to large towns. These estimates were produced by creating a conductance surface by assigning travel speeds to each pixel based on the national road network and land cover and calculating the quickest travel time to each market using least-cost-path algorithms. The road network in Tanzania was obtained from Open Street Maps and travel speeds were assigned depending on their classification (primary, secondary or tertiary highways). We assigned travel speeds to the pixels outside the road network by using the Globcover 2009 Version 2.3 Land Cover Classification [40] and assuming different travel speeds in each land cover class. With the travel speed covering Tanzania, market access was estimated by calculating the least accumulated time from each pixel to a town with a population of more than 50,000 inhabitants. Market town locations were taken from the GRUMP database [41].

### Fertilizer scenarios

We evaluate fertilizer profitability over four different application rate scenarios: no nitrogen usage (ZERO), a commonly recommended N application rate of 55 kg/ha (BK, [42]) and two potential fertilizer recommendations optimized to obtain the highest maize yields (OPyield) or the highest net revenue (OPnetrev) obtainable at any particular location. Optimization scenarios use the average seasonal rainfall data over the 1980–2019 period as the basis for the calculation.

## Results

### Market and farm-gate prices of maize and fertilizer

Maize market prices in the LSMS dataset had a mean of 0.37 USD/kg and ranged from 0.07 to 0.94 USD/kg. The random forest model performed well, explaining 92% of this variation of the training data with an RMSE = 0.17. Predicted prices ranged from 0.14 USD/kg in the Southern Highlands and areas near Shinyanga to a maximum of 2 USD/kg in some western regions (Fig 2A). After accounting for transport costs from local markets, 60% of the territory had predicted farm-gate between 0.2 and 0.4 USD/kg (Fig 2B). Prices reach to a maximum of 0.69 USD/kg in Zanzibar. Areas with large maize production, such as the Southern Highlands and areas near Arusha, had predicted farm-gate prices near 0.3 USD/kg. 12% of Tanzania is predicted to have no positive farm-gate maize price.

Because of the sparse number of large market towns, our model predicts that approximately 77% of the territory has a nitrogen price of above 2.5 USD/kg and only 12% of the area with prices lower than 1.5 USD/kg (Fig B in S1 Text). Only northern and eastern regions, with a higher density of cities and roads, present large areas with lower fertilizer prices.

### Yield response model

The yield response model fitted with the random forest model explained 24% of the variance found in the yields reported in the APS household survey (Fig 3A). The applied nitrogen rate was the variable with the highest importance followed by rainfall, altitude and soil organic carbon (Fig A in S1 Text). Crop management variables such as improved seed, manure use, intercropping were useful predictors in the model.

Partial dependency plots for the random forest model showed a positive response of yield to rainfall, nitrogen and soil organic carbon (Fig 3B–3D). Increasing seasonal rainfall between the 500 mm and 1250 mm positively affects yields, as we would expect. (Negative effects of

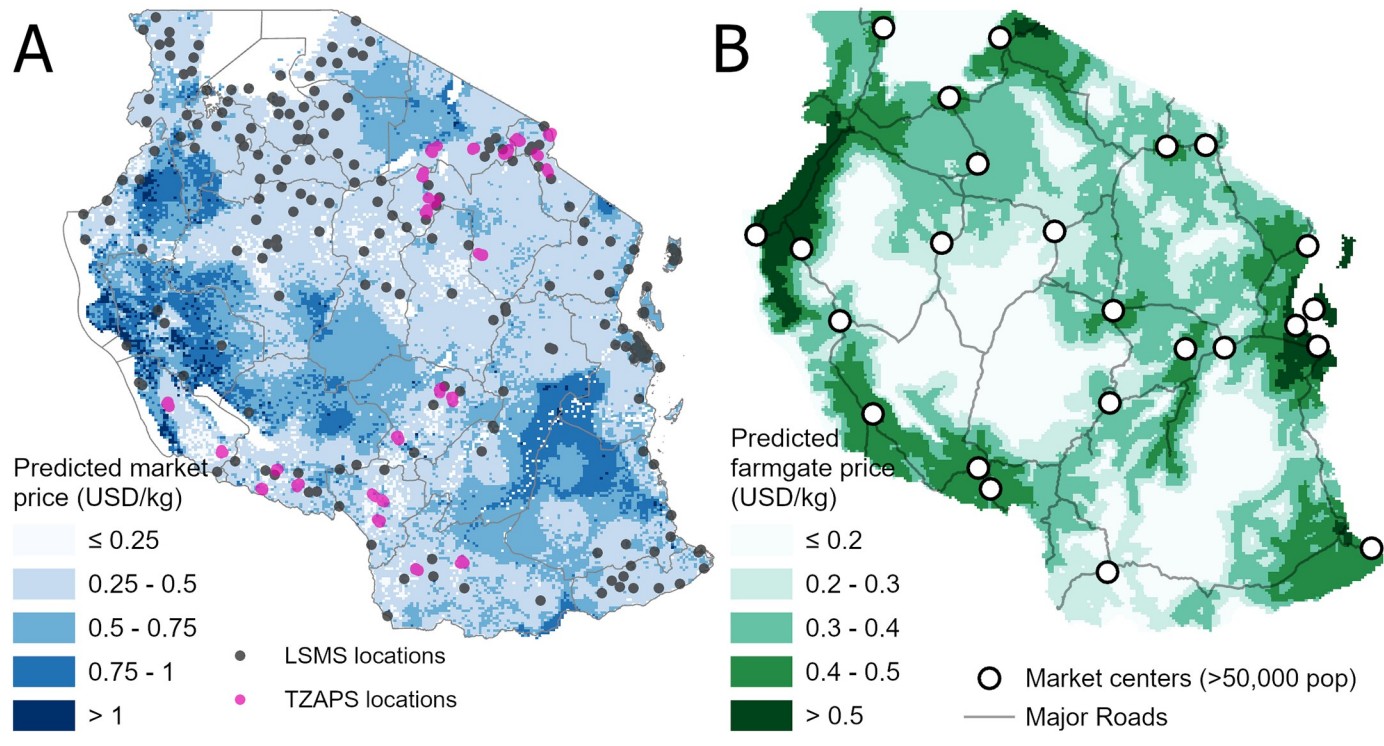

**Fig 2. Predicted maize prices in Tanzania.** (A) Market prices. (B) Farm-gate prices.

increasing precipitation were predicted between 0 mm to 500 mm, although very few sample points fall in this rainfall range; these results may reflect unobserved irrigation practices.) Yield increases rapidly when increasing the amount of nitrogen with diminishing returns after an application of 300 kg/ha.

## Simulation results

Given spatial price variability, higher yields are not necessarily associated with higher net revenues (Fig C in S1 Text). In our scenarios, a blanket recommendation will result, on average, 27% more production and a -4% increase in returns compared to a baseline scenario across the country. Optimizing fertilizer to maximize yields results in an increase of 57% of yields and a 10% reduction in profitability. Optimizing for returns results in a significant increase in yields, 47% more than without the use of nitrogen, but an average increase of 16% of returns across the maize distribution.

However, these yield and profitability changes are highly dependent on the region of the country. Yields changes vary with soil conditions, while profitability is more dependent on the proximity to large market towns. According to our model, the maximum yields are obtained with an application of 175 kg/ha of N for over 90% of the maize distribution and can increase the production over 60% in areas such as the southern highlands and Dodoma and Singida, but decrease the profitability because of the low farm-gate prices. Maximizing for yields is predicted to most profitable in areas such as Dar es Salaam, Musoma, Mwanza and Kigoma, where the farm-gate maize price is high enough to allow higher investments in nitrogen.

Optimizing for higher yields by using the highest possible N dose can also result in prohibitive costs of fertilizers that are not recovered with the sale of the higher production, resulting in negative net revenues. Based on our fertilizer price model, areas such as Rungwa and the

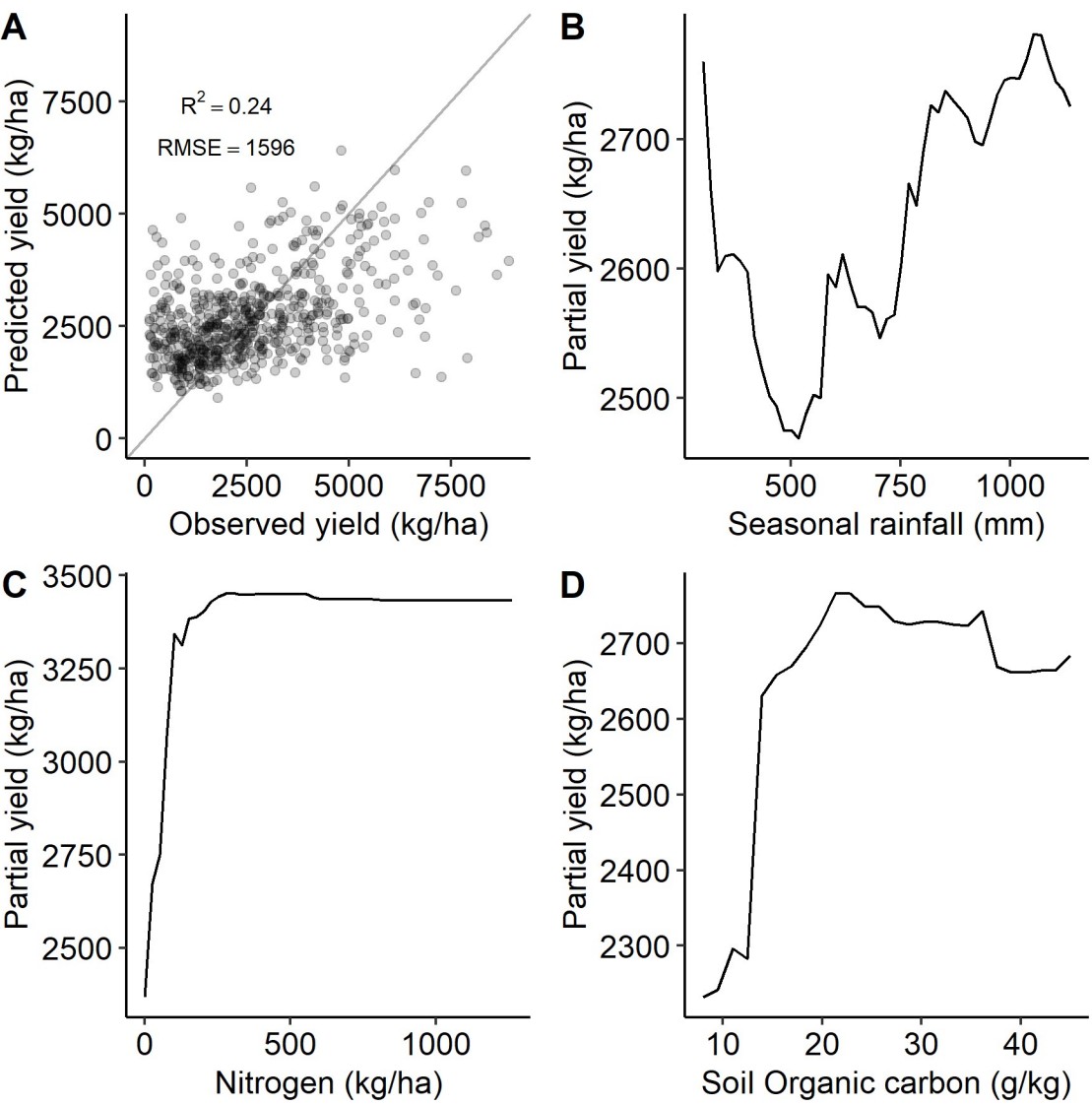

**Fig 3. Yield response random forest model selected results.** (A) Observed vs predicted yield and fitness measures of the yield model. Partial dependence plots of (B) seasonal rainfall, (C) nitrogen and (D) organic carbon from the yield random forest model.

inland regions of the southeast are predicted to perform better with no fertilizer. Only a substantial decrease in fertilizer prices, most likely as a result of higher accessibility, can improve the profitability of nitrogen applications in these areas.

Maximum profitability is a result of only moderate increases in yields, especially in northwestern regions. High nitrogen rates that maximize net revenue are mostly correlated with high accessibility areas (Fig 4). Only crop areas near cities have recommendations above 125 kg/ha and only 6% of the maize distribution may benefit from applications above 100 kg/ha. Recommendations between 50 and 100 kg/ha are estimated to be appropriate for 75% of the territory. In the simulation results, areas near large market towns were those which most benefitted economically from the use of nitrogen (Table 2). Regions such as Dar es Salaam, Kigoma and Mara, with Mwanza, Kagera and Kigoma, with large rural populations and high

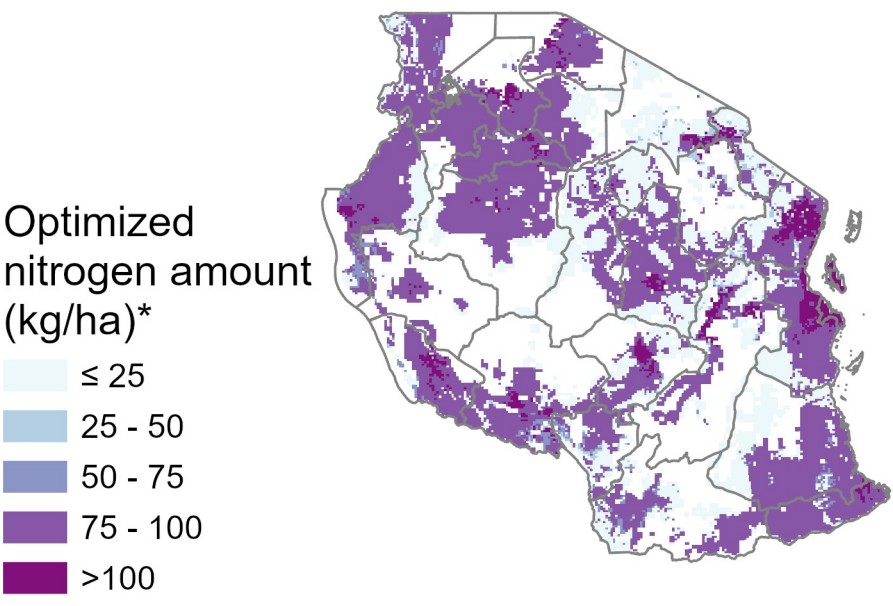

*To obtain maximum profitability (scenario: OPnetrev)

**Fig 4. Optimized amounts of nitrogen fertilizer rate to maximize net revenue.**

maize farm-gate prices, are predicted to have the highest returns to nitrogen fertilizer investments.

The distribution of the potential profitability distribution in each location was calculated accounting for pixel level rainfall variability. Results are shown in Fig 5B (with rainfall variability expressed as coefficient of variation in panel A for reference). Over 44% of the crop distribution has a coefficient of variation higher than 5% when using an optimized nitrogen rate and accounting for seasonal rainfall variability. The lower predicted production in the northern and northeastern regions results in higher uncertainty in the distribution of returns. The Southern Highlands has lower pixel rainfall variability and, combined with the higher yield results, relatively high expected returns with low variability.

## Validation

As a way of validating whether or not predicted profitability has any practical value, we used the log of predicted net revenue from our baseline scenario, as well as the standard deviation of net revenue as a measure of uncertainty, in a model of fertilizer usage by smallholder farmers in Tanzania, using three panel waves of the Tanzania LSMS National Panel Survey (2008/9, 2010/11, and 2012/13). For ease of interpretation, we use a linear probability model. To address time-invariant unobserved heterogeneity which might otherwise bias our results, we modeled the unobserved time invariant heterogeneity as a function of the time-averages of (time-varying) observed characteristics (i.e. the Mundlak-Chamberlain device [43, 44]). Thus, time-averages are added to the model, but not interpreted. Results, shown in Table 3, indicate that the log of expected profitability is a strong positive correlate of fertilizer usage, and the standard deviation of expected profitability is a strongly negative correlate of fertilizer usage. The latter result is consistent with stylized empirical finding African smallholders are less likely to make fertilizer investments if the returns have higher levels of uncertainty. The fact that these predicted profitability indicators are significant correlates even after controlling for

**Table 2. Summary table of aggregate gains in net revenue.**

| Region | Rural population (million) | Maize area (km2) | Average gains when changing nitrogen scenarios (USD/ha) | |
|---|---|---|---|---|
| | | | ZERO to OPnetrev | BK to OPnetrev |
| Arusha | 1.7 | 723.9 | 23.5 | 118.2 |
| Dar es Salaam | 0.3 | 13.1 | 463.9 | 297.9 |
| Dodoma | 2.6 | 933.8 | 68.6 | 106.2 |
| Geita | 1.8 | 1085 | 279.9 | 147.8 |
| Iringa | 1.1 | 1444.2 | 107.8 | 90.6 |
| Kagera | 3.3 | 702.3 | 225.1 | 99 |
| Katavi | 0.7 | 583.9 | 219.2 | 101.4 |
| Kigoma | 2.2 | 1296.8 | 369.8 | 158.5 |
| Kilimanjaro | 1.8 | 627.4 | 31.7 | 79.1 |
| Lindi | 1.1 | 482.7 | 108 | 86.6 |
| Manyara | 2.1 | 1348.8 | 20 | 88 |
| Mara | 2.2 | 491.5 | 374.6 | 199.7 |
| Mbeya | 3.0 | 2463.5 | 165 | 107.3 |
| Morogoro | 2.2 | 1248.5 | 78.9 | 79 |
| Mtwara | 1.5 | 688.5 | 241.4 | 126.8 |
| Mwanza | 2.9 | 689.9 | 271.5 | 178.2 |
| Njombe | 0.9 | 860.8 | 154.8 | 48.6 |
| Pwani | 1.2 | 633 | 150.8 | 120.5 |
| Rukwa | 1.2 | 841.6 | 335.2 | 139.6 |
| Ruvuma | 1.6 | 959.9 | 88.9 | 60.8 |
| Shinyanga | 2.2 | 942.3 | 178.1 | 134.7 |
| Simiyu | 2.3 | 1234.2 | 80.1 | 103.6 |
| Singida | 1.7 | 728.3 | 28.9 | 76.8 |
| Tabora | 3.0 | 1639.5 | 136.7 | 107.2 |
| Tanga | 2.3 | 1492 | 112.7 | 87.4 |

region and travel time from each household location to the nearest market town suggests that there is information content in our model predictions beyond simply proxying for market remoteness.

## Robustness check

Our estimated agronomic use efficiencies (AE) are low compared with reported values from researcher managed studies [32, 45]. We attribute this to the observational nature of our data: as others have noted, calculated nitrogen use efficiencies for maize are much higher on researcher-managed plots than on plots managed exclusively by smallholders [45–47]. Our estimated use efficiencies (mean of 7.2 kg grain per additional kg of N) are comparable to those found by [21] using LSMS-ISA data for Tanzania (7-8kg). Nonetheless, as a robustness check, we re-estimate the predicted spatial distributions of fertilizer profitability under assumptions of 125% and 150% increases in our predicted agronomic use efficiency (bringing the mean value from 7.2 to 9.2 and 11 kg/kg, respectively. The resulting changes to the cumulative distribution of profitability (Fig 6) are relatively modest: when moving from our estimated agronomic efficiency distribution (mean = 7.2 kg/kg) to a distribution with a mean of 9.2 kg/kg (i.e., 125% of the AE predicted by our model), we have an increase of 3% of pixels for which fertilizer net revenue exceeds 100 USD/ha (i.e., from 90% to 93%). When we assume an agronomic efficiency distribution with a mean of 11 kg/kg (i.e., 150% of our predicted AE), we

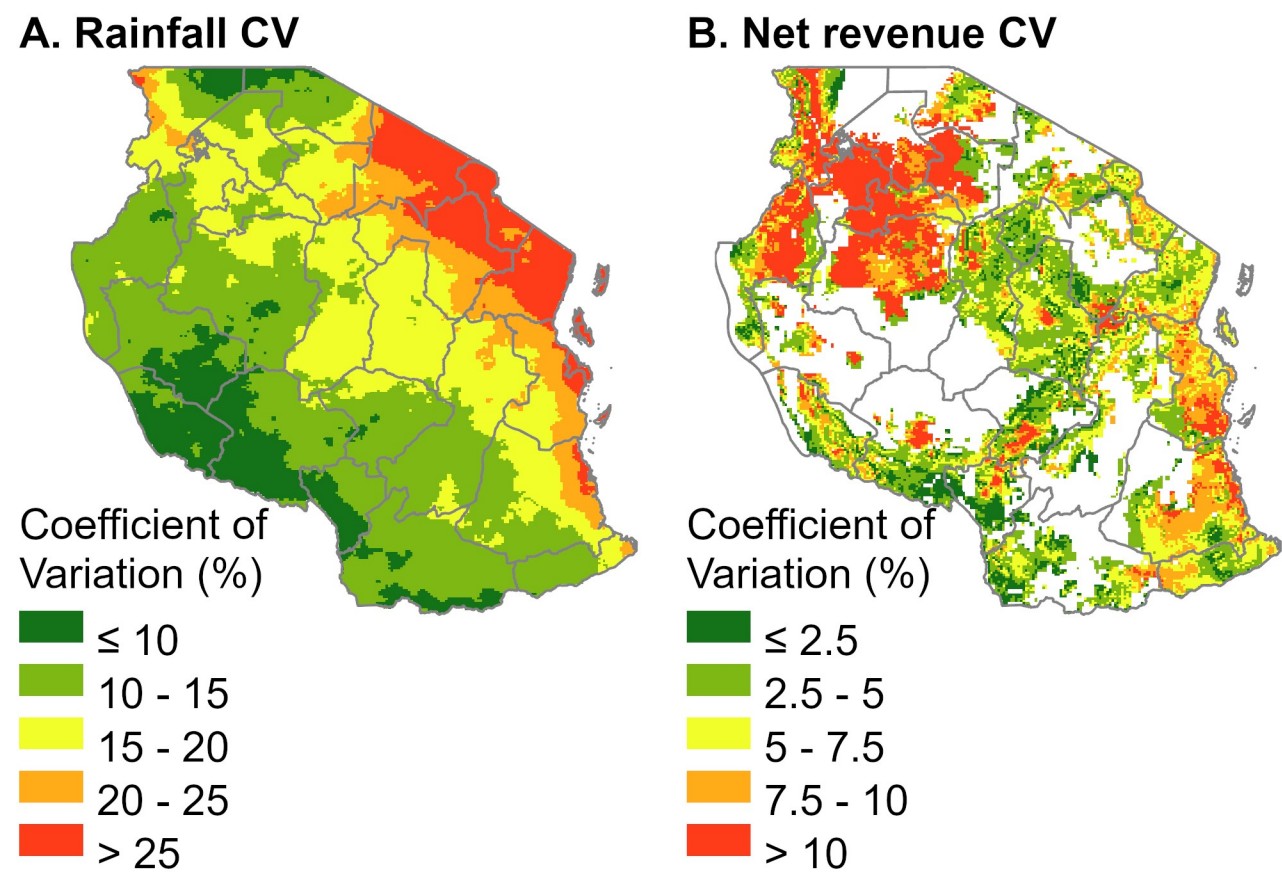

**Fig 5. Rainfall and net revenue variation.** (A) Seasonal (December-May) rainfall coefficient of variation. (B). Net revenue coefficient of variation resulting from the OPnetrev scenario.

have an increase of 4% of pixels for which fertilizer net revenue exceeds 100 USD/ha (i.e., from 90% to 94%).

## Discussion and conclusions

This paper has illustrated a simple yet useful method of predicting yield responses to fertilizer over heterogeneous production landscapes, with a view toward guiding strategic investments and policy interventions. In our case study of smallholder Tanzanian maize farmers, our results indicate highly variable fertilizer responses over geographic space, in line with other empirical studies in the region (e.g., [4–7]). While fertilizer use is profitable, on average, it is not profitable everywhere. Farmers in very remote areas would not gain financially, given local input-output price ratios, even where agronomic returns are high. This result underscores the importance of acknowledging spatial differences in economic remoteness, and the implications for technology profitability, in designing agronomic interventions. This is particularly important for primarily agrarian economies with large shares of the rural population in remote areas, conditions that characterize many of the countries in sub-Saharan Africa.

Our analysis has direct implications for the debate on closing yield gaps. Closing yield gaps may not be economically feasible in areas which are remote from markets. However, using a framework such as the one we propose may help to identify where to prioritize investments in closing yield gaps, i.e., where returns on investment are largest.

**Table 3. Validation: Out of sample prediction of fertilizer usage.**

| Dep var: fertilizer user (=1) | (1) | (2) |
|---|---|---|
| log(net revenue) | 0.0952*** | 0.110*** |
| | (2.84e-06) | (1.76e-07) |
| std.dev.(net revenue) | | -0.000691*** |
| | | (0.00101) |
| area cultivated | 0.000498 | 0.000466 |
| | (0.771) | (0.787) |
| age of head | -0.000105 | -0.000168 |
| | (0.770) | (0.639) |
| female head (=1) | -0.00568 | -0.00662 |
| | (0.668) | (0.617) |
| education of head | 0.00985*** | 0.00960*** |
| | (8.46e-09) | (1.76e-08) |
| # members | 0.000830 | 0.00100 |
| | (0.777) | (0.732) |
| log value of productive assets | 0.00621*** | 0.00635*** |
| | (0.00681) | (0.00576) |
| log travel time to market town | -0.0372*** | -0.0382*** |
| | (1.82e-05) | (1.04e-05) |
| mean annual rainfall 1997–2014 | 0.000113** | 7.99e-05 |
| | (0.0349) | (0.143) |
| Region FE? | yes | yes |
| Year FE? | yes | yes |
| Mundlak-Chamberlain device? | yes | yes |
| Observations | 5,819 | 5,819 |
| R-squared | 0.236 | 0.238 |

Dependent variable is a dummy indicator taking a value of 1 if the household is a user of inorganic fertilizer. Data are from the 2009, 2010 and 2013 waves of the Tanzania LSMS-ISA data, restricted to landholding households in the rural areas. Standard errors are robust to clustering at the enumeration area level. Model (2) includes the standard deviation of the expected profitability.

Our results also highlight the importance of acknowledging uncertainty in modeling the returns to investments from a farmer's perspective. In our modeling framework, both agronomic and economic returns to fertilizer investments are strongly conditioned by location-specific rainfall patterns. Given the uncertainty around rainfall outcomes in any particular year, there is a corresponding amount of uncertainty in investment returns from the farmer's perspective. The variability of estimated returns from our model is a good predictor of actual fertilizer usage in a nationally representative sample of Tanzanian smallholders: higher levels of uncertainty around investment returns are strongly negatively associated with the likelihood of fertilizer usage. This finding underscores the role that risk plays risk in smallholder decision-making and signals that technology promotion efforts, which fail to address economic risk, are fundamentally flawed. In addition, and, conversely, agronomic practices that improve agronomic use efficiencies through technology promotion can reduce these financial risks.

Only when farmer decision-making is more fully integrated into planning frameworks will the required changes in production technology begin to take place at the scale necessary to deliver expected changes in smallholder-dominated food systems. The *ex ante* framework articulated in this paper is one way in which economic returns and the variability of those

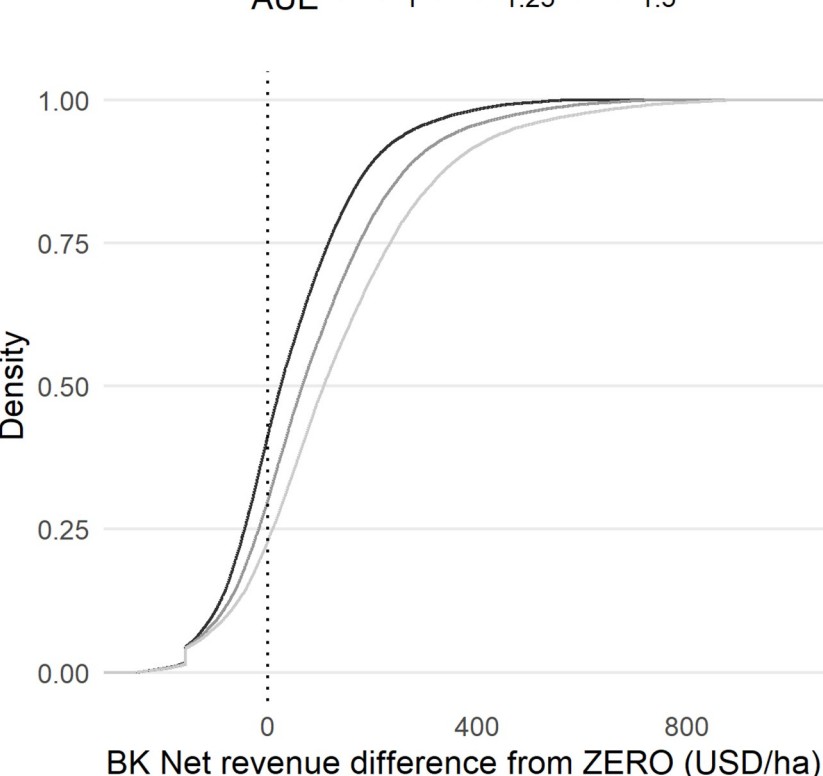

**Fig 6. Cumulative distribution of net revenue differences of the BK scenario from the ZERO scenario under different agronomic use efficiencies (AUE) assumptions.**

returns may be better linked with agronomic modeling and incorporated into strategic planning and targeting frameworks. We envision such frameworks becoming increasingly important ways to address the challenge of increasing sustainable intensification efforts in the region.

## Supporting information

**S1 Text.**
(DOCX)

## Author Contributions

**Conceptualization:** Sebastian Palmas, Jordan Chamberlin.

**Formal analysis:** Sebastian Palmas, Jordan Chamberlin.

**Funding acquisition:** Jordan Chamberlin.

**Investigation:** Sebastian Palmas.

**Methodology:** Sebastian Palmas, Jordan Chamberlin.

**Supervision:** Jordan Chamberlin.

**Visualization:** Sebastian Palmas.

**Writing – original draft:** Sebastian Palmas, Jordan Chamberlin.

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
