## [Decision Letter · Decision Letter 0]

30 Jun 2020

PONE-D-20-10700

Fertilizer profitability for smallholder maize farmers in Tanzania: A spatially-explicit ex ante analysis

PLOS ONE

Dear Dr. Palmas,

Thank you for submitting your manuscript to PLOS ONE. After careful consideration, we feel that it has merit but does not fully meet PLOS ONE’s publication criteria as it currently stands. Therefore, we invite you to submit a revised version of the manuscript that addresses the points raised during the review process.

We look forward to receiving your revised manuscript.

Kind regards,

Luigi Cembalo, PhD

Academic Editor

PLOS ONE

Journal Requirements:

Reviewers' comments:

Reviewer's Responses to Questions

**Comments to the Author**

1. Is the manuscript technically sound, and do the data support the conclusions?

Reviewer #1: Partly

Reviewer #2: Yes

2. Has the statistical analysis been performed appropriately and rigorously? 

Reviewer #1: Yes

Reviewer #2: Yes

3. Have the authors made all data underlying the findings in their manuscript fully available?

Reviewer #1: Yes

Reviewer #2: Yes

4. Is the manuscript presented in an intelligible fashion and written in standard English?

Reviewer #1: Yes

Reviewer #2: Yes

5. Review Comments to the Author

Reviewer #1: Dear authors,

It was a pleasure to read the well written manuscript. The issues around fertilizer profitability and risk are important for both smallholders and governments with regional implications for food security.

The approach is highly interesting, accounting for location in prices and crop yield responses combined with temporal variability due to rainfall. I do have some concerns about the approach used with respect to the grain yield response to N applications and increasing rainfall.

1. The key component is the yield response to N-application determining the net return of investment in fertilizers. I would appreciate if authors can consistently use N application rather than fertilizer as that may refer to types of fertilizer with very different N concentrations. For example, the national voucher scheme uses 2 vouchers: one for 50 kg DAP or 50 kg of rock phosphate and one for 50 kg Urea or for ammonium sulphate. This equates to 100 kg fertilizer, but to only 15*0.18 + 50*0.46 = 32 kg N for DAP + Urea but also to 0 + 50*0.21 = 10.5 kg N when rock phosphate is combined with ammonium sulphate. The latter with be very rare. I would not expect that these amounts are applied on exactly one ha, which seems to be suggested by the authors (line 188), with reference to a World bank report that does not detail recommendations. Typically, recommendations are 50 kg of DAP + 50 kg of CAN per acre for e.g. OneAcreFund clients, amounting to 55 kg N/ha.

2. A pure empirical approach to estimate yield responses to fertilizer is somewhat problematic. The data given in Figure 3c suggest that a 125 kg N / ha application increases yields by about 900 kg/ha, reflecting an agronomic efficiency (AE) of 900/125 = 7.2 kg grain / kg N applied. This is a very low value, much below values found for agronomic experiments (25-50 kg/kg, depending on PK fertilization, Ichami et al 2020)) on-farm trials that typically are around 10-11 kg/kg across a range of environments (Rurinda et al., 2020). This suggest that other growth limitations played an important role, e.g. water deficiency and pests and diseases that reduced growth. Some additional context about these seasons would be much appreciated as that helps to interpret the outcomes.

3. The response to N strongly depends on applications of P and K. In most areas of Tanzania, k will not be very problematic. However, the response to N is strongly affected by availability of P in the soil and use of P fertilizers. Surprisingly, P application is not included as a co-variate, although it strongly affects the price of fertilizers used. What prices were used for the calculations, was this including P?

4. Authors included rainfall data from 2 years in their model, and variation in rainfall is spatial. These spatial correlations cannot be transferred to temporal variations. Farmers adapt their management to an expected yield and variability: in areas with consistently good seasons input levels will be higher and management geared towards high yields; in areas with a higher risk of droughts, lower inputs will be used resulting in lower maximum yields even when rainfall is abundant. Translating rainfall variability across years to net revenue based on a “spatial” model wihout a proper temporal component is in my view flawed as it assumes that farmers are ignorant for this temporal variability. They obviously are not and have a range of management adaptations to droughts and risks. This should be properly discussed.

Points for clarification

Authors mention that crop cuts were taken. Are grain yields referring to dry matter yields or “ fresh” yields? Is this for grain or for grain + cobs? How was fertilizer use measured on the plots measured? Farmer reported inputs are very uncertain too. Please provide required details. The range of N applications seems near impossible: intentional applications of more than 150 kg/ha are very rare in SSA.

Line 37: remove “once considering the local farm-gate crop and fertilizer prices”.

Line 52: risk-reducing rather than risk-smoothing. Smoothing is a data operation and not a financial instrument.

Liners 66-80: this part is odd: is more a summary than an intro for the approach and what can be expected in the manuscript.

Line 110: refrase sentence. Something like: A third key conceptual feature of our approach is based on accounting for the stochasticity of responses.

Table 1. Header: add meaning of SD. Maybe add how household size is measured for clarity. Can you explain why an average value for a boolean variable can be >1? This applies to female heads of households.

Line 168: is this per kg of N or per kg of fertilizer, and if so of what type?

Line 188: the reference does not detail recommendations, but explains voucher schemes. See point 1. Typical recommendations are 55 kg N/ha, including 1 bag of CAN + 1 DAP per acre.

Line 199: are values per kg of grain at 12.5%moisture?

Line 201: RMSE value seems impossible given the ranges in prices. Please add a unit to the corrected number.

Line 217: give full abbreviation of TZAPS when first used please.

Line 235: explain what is 1% referring to after “an average increase of 1% across...”

Line 239, 242 and 245: please be consistent: aren’t authors just optimizing for yields? Maximum yields would just require highest possible dose...

Line 265: authors mean coefficient of variation >5% rather than profitability (not shown in Figure 5).

Line 276-277: puzzling sentence. Did authors add the averages of regressors that vary across years or did authors replace those variables by the average values? Time invariant unobserved heterogeneity is boiling down to spatial variability or consistent differences in management practices....maybe just mention that? How does this affect rainfall and associated interactions?

Line 280-282: refrase this sentence please.

Line 314-316: or emphasises that technology promotion needs to be embedded in good agronomic practises. From the results, technological risks (i.e. low AEs) are dominating and translating into financial risks, but these should be dampened by good practises which is very feasible.

References:

Ichami, S.M., Shepherd, K.D., Sila, A.M., Stoorvogel, J.J., Hoffland, E., 2019. Fertilizer response and nitrogen use efficiency in African smallholder maize farms. Nutr Cycl Agroecosyst 113.

Rurinda, J., Zingore, S., Jibrin, J.M., Balemi, T., Masuki, K., Andersson, J.A., Pampolino, M.F., Mohammed, I., Mutegi, J., Kamara, A.Y., Vanlauwe, B., Craufurd, P.Q., 2020. Science-based decision support for formulating crop fertilizer recommendations in sub-Saharan Africa. Agric. Sys. 180, 102790.

Reviewer #2: This paper presents a methodological framework for addressing the question of how to target interventions to enhance the benefits of using inorganic fertilizer under high spatial heterogeneity and rainfall variability faced by smallholder farmers. The framework addresses this question by estimating the response of yields and profitability to inorganic fertilizer application using location-specific data in the case of maize in Tanzania by combining different sources of data from household surveys to spatial data. The authors use methods that I have never seen applied to issues of agronomic responses before, such as the Random Forest model. In fact, I had to find out exactly what this model entailed, so by reviewing this paper I have learned something completely new to me. In my opinion, this is a very interesting and innovative paper. It is well-written and useful. I have however a few comments.

The authors mention three constraints for low fertilize usage: gap in agronomic response, low and variable economic returns, and risk averseness. However, a constraint that is not explicitly addressed is the lack of physical availability. Even if a farmer wants to buy fertilizer, nobody may sell it in the area. Obviously, this means a very high price due to transportation costs, but also search costs. So fertilizer is expensive and difficult to find and procure. In fact, one of the reasons for outside interventions is to increase the physical availability of fertilizer. So, the constraints the authors mention are demand constraints, but there they fail to mention supply constraints.

In lines 46-48, the authors state that there has been a dearth of planning and targeting frameworks such as the one they propose in the paper. This is a very important point, but it may be useful to discuss whether there have been other types of frameworks to address the issue of targeting interventions to increase access to fertilizer.

Also, it may be useful as part of the background to talk about programs to procure fertilizers to farmers. No mention of those explicitly. Particularly in the context of Tanzania.

Lines 13- 131. Fertilizer response was modeled with ~ 14.5 farms per district. Is this enough to capture the variability present in Tanzania?

Sine I assume that many of the readers of this paper may be like myself, i.e. not familiar with Random Forest model, I suggest, if possible, that the authors refer to a paper that provides a non-technical explanation of the model. They do cite the paper by Breiman L., but it is quite technical. I had to search the Internet to find information on the model. Providing a non-technical reference to the model will be very useful for a reader not familiar with the model.

Table 1 shows that there were 601 pooled observations in the farm survey, based on 362 farms for two years. Since there where 362 households, this indicates that there were 123 missing cases (362*2= 724-601). They do not mention that and the reasons for missing data.

In the same table, for the variable female head of household (yes=1) it states a value of 7.067. This value does not make sense. I imagine it refers to 7%. So for consistency should be expressed as 0.07

Lines 152-154, could you give a short summary of the approach used. Not everybody has the time to read the original paper. I extracted this from the paper they cite:

“We show that in many countries this variation can be predicted for unsampled locations by fitting models of prices as a function of longitude, latitude, and additional predictor variables that capture aspects of market access, demand and environmental conditions.”

Lines 156-160 summarizes the method to estimating farm-gate maize prices. Refers to Table A in SI text, lots of variables used. What was the number of observations of prices used?

Lines 278-280 and Table 3 be more explicit about what (1) and (2) mean, I assume that in (1) std dev is not considered and in (2) it is. Please clarify.

I think that the authors could make the paper more interesting if they include in the discussion section their views on the implications of their findings for the debate on closing yield gaps. This is an important issue in the literature, and I see that their results are very relevant. They show that in fact maximizing yield is not profitable and that what is profitable will translate into maintaining a relatively large yield gap. Worth discussing.

In general, I think this is an very good paper that deserves to be published.

6. PLOS authors have the option to publish the peer review history of their article (what does this mean?). If published, this will include your full peer review and any attached files.

Reviewer #1: No

Reviewer #2: No

---

## [Author Response · Author response to Decision Letter 0]

13 Aug 2020

Reviewer #1: 

Dear authors,

It was a pleasure to read the well written manuscript. The issues around fertilizer profitability and risk are important for both smallholders and governments with regional implications for food security.

The approach is highly interesting, accounting for location in prices and crop yield responses combined with temporal variability due to rainfall. I do have some concerns about the approach used with respect to the grain yield response to N applications and increasing rainfall.

We thank the reviewer for this positive overall assessment of our paper, and for the specific comments and suggestions, which we found very helpful. We have addressed each of these comments in detail below.

1. The key component is the yield response to N-application determining the net return of investment in fertilizers. I would appreciate if authors can consistently use N application rather than fertilizer as that may refer to types of fertilizer with very different N concentrations. For example, the national voucher scheme uses 2 vouchers: one for 50 kg DAP or 50 kg of rock phosphate and one for 50 kg Urea or for ammonium sulphate. This equates to 100 kg fertilizer, but to only 15*0.18 + 50*0.46 = 32 kg N for DAP + Urea but also to 0 + 50*0.21 = 10.5 kg N when rock phosphate is combined with ammonium sulphate. The latter with be very rare. I would not expect that these amounts are applied on exactly one ha, which seems to be suggested by the authors (line 188), with reference to a World bank report that does not detail recommendations. Typically, recommendations are 50 kg of DAP + 50 kg of CAN per acre for e.g. OneAcreFund clients, amounting to 55 kg N/ha.

We appreciate the need for clarity that the reviewer notes. We have revised the manuscript to consistently refer to N application rather than fertilizer, wherever this makes contextual sense.

We also revised our blanket recommendation assumption to a rate of 55 kg/ha, is in line with the reviewer’s suggestion, which indeed corresponds to the One Acre Fund recommendation. We now note this on page 11 of the revised manuscript. We changed the reference from the World Bank report to the work by Kanyeka et al, which includes the recommendation amount used. The N content of CAN depends on the brand, but if we use 27%, then the OAF recommendation is: ((.18*50)+(.27*50))/0.404686 = 55.6 kg/ha

2. A pure empirical approach to estimate yield responses to fertilizer is somewhat problematic. The data given in Figure 3c suggest that a 125 kg N / ha application increases yields by about 900 kg/ha, reflecting an agronomic efficiency (AE) of 900/125 = 7.2 kg grain / kg N applied. This is a very low value, much below values found for agronomic experiments (25-50 kg/kg, depending on PK fertilization, Ichami et al 2020)) on-farm trials that typically are around 10-11 kg/kg across a range of environments (Rurinda et al., 2020). This suggest that other growth limitations played an important role, e.g. water deficiency and pests and diseases that reduced growth. Some additional context about these seasons would be much appreciated as that helps to interpret the outcomes.

This is a fair point, and deserves further elaboration. Trials on farmer’s fields, while more representative of actual smallholder farm responses than researcher-managed agronomic experiments, are still often managed with management protocols (and very small plot sizes) that result in yields which are higher on average than those found in observational data, i.e. farm survey data. As an example, Mather et al. (2016) found for Tanzania using LSMS-ISA data (7-8kg). Rurinda et al 2020 uses nutrient omission trials data, which are on farmer’s fields, but follow optimal management protocols. As such, we would expect them to be a bit higher. 

However, in the current version of the paper, we have included two additional assumptions: where estimated AE values are increased to 125% and 150% of the empirically estimated rates, bringing the average AE up to 9.2 and 11, respectively. This sensitivity analysis is described on page 18 of the revised manuscript.

3. The response to N strongly depends on applications of P and K. In most areas of Tanzania, k will not be very problematic. However, the response to N is strongly affected by availability of P in the soil and use of P fertilizers. Surprisingly, P application is not included as a co-variate, although it strongly affects the price of fertilizers used. What prices were used for the calculations, was this including P?

In order to keep the analysis simple, we focus on N, accounting for the cheapest source of N in our pricing. We use spatially distributed market prices for urea as reported in the LSMS data to predict the spatial distribution of nitrogen. P is excluded from this price calculation for reasons of simplicity (although it is included in the yield response random forest model). As such, our pricing and profitability estimates can be taken as lower bounds. 

4. Authors included rainfall data from 2 years in their model, and variation in rainfall is spatial. These spatial correlations cannot be transferred to temporal variations. Farmers adapt their management to an expected yield and variability: in areas with consistently good seasons input levels will be higher and management geared towards high yields; in areas with a higher risk of droughts, lower inputs will be used resulting in lower maximum yields even when rainfall is abundant. Translating rainfall variability across years to net revenue based on a “spatial” model wihout a proper temporal component is in my view flawed as it assumes that farmers are ignorant for this temporal variability. They obviously are not and have a range of management adaptations to droughts and risks. This should be properly discussed.

To clarify, we use a pixel specific measure of rainfall variability in our simulation. Thus, the temporal variability we account for in our estimation of uncertainty of returns is location-specific. We have specified this in the text on page 11 of the revised manuscript, which now reads as follows: “Accounting for rainfall variability will gives us an insight into the distribution of the profitability in each location. To account for seasonal rainfall variability, we run each scenario using 21 years of spatial time-series estimates of seasonal rainfall for 21 years of rainfall data between 1980 1987 andto 2019 from the TAMSAT v3.0 dataset [36]. We calculate the pixel level distribution of the expected yield and net revenue results on the basis of the resulting model output from each seasonal rainfall estimate (holding other spatial covariates at their observed temporally-invariant levels).”

Points for clarification

Authors mention that crop cuts were taken. Are grain yields referring to dry matter yields or “ fresh” yields? Is this for grain or for grain + cobs? How was fertilizer use measured on the plots measured? Farmer reported inputs are very uncertain too. Please provide required details. The range of N applications seems near impossible: intentional applications of more than 150 kg/ha are very rare in SSA.

The crop cut protocol involved collecting a grain sample, which was dried to 15% moisture before weighing. This is the same protocol used by Rurinda et al. 2020. - We have clarified this in the revised manuscript on page 7.

An aggregate N application rate was calculated on the basis of all the fertilizer applications reported by the farmer for that field – i.e. recorded across multiple fertilizer types and application rates, and normalized by the size of the field. The higher end of the application rate generally corresponds to very small plots. We have clarified this in the text on page 7.

Line 37: remove “once considering the local farm-gate crop and fertilizer prices”.

While we are aware that these words may be redundant, we think they serve to clarify and emphasize our transformation of agronomic returns into economic returns, thus increasing clarity of the analysis.

Line 52: risk-reducing rather than risk-smoothing. Smoothing is a data operation and not a financial instrument.

Changed to “risk-reducing”

Liners 66-80: this part is odd: is more a summary than an intro for the approach and what can be expected in the manuscript.

Yes, this indeed is a summary statement of the results which are explained in more detail in the body of the paper.

Line 110: refrase sentence. Something like: A third key conceptual feature of our approach is based on accounting for the stochasticity of responses.

We have rephrased as suggested on page 3

Table 1. Header: add meaning of SD. Maybe add how household size is measured for clarity. Can you explain why an average value for a boolean variable can be >1? This applies to female heads of households.

We have updated the table title. Household size definition added. “Female heads of households”was incorrect, changed to “Years of education of head of household:

Line 168: is this per kg of N or per kg of fertilizer, and if so of what type?

We have now specified that it is kg of N on page 10.

Line 188: the reference does not detail recommendations, but explains voucher schemes. See point 1. Typical recommendations are 55 kg N/ha, including 1 bag of CAN + 1 DAP per acre.

We also reviewed the blanket recommendation and changed it to a rate of 55 kg/ha.

Line 199: are values per kg of grain at 12.5%moisture?

The crop cut protocol involved collecting a grain sample, which was dried to 15% moisture before weighing. This is the same protocol used by Rurinda et al. 2020. - We have clarified this in the revised manuscript on page 7.

Line 201: RMSE value seems impossible given the ranges in prices. Please add a unit to the corrected number.

Corrected. The error was because RMSE was being calculated using Ethiopian Birr instead of USD.

Line 217: give full abbreviation of TZAPS when first used please.

Updated and full abbreviation given in Line 131.

Line 235: explain what is 1% referring to after “an average increase of 1% across...”

Specified in page 13 that it is a % increase of returns.

Line 239, 242 and 245: please be consistent: aren’t authors just optimizing for yields? Maximum yields would just require highest possible dose...

In Figure 3 we show that the random forest model predicts diminishing returns with high concentrations of N when maintaining every other covariate constant. Our model suggests that the highest possible yields are obtained with an N application of 175 kg/ha. We have clarified this in the revised manuscript on page 13.

Line 265: authors mean coefficient of variation >5% rather than profitability (not shown in Figure 5).

Corrected to coefficient of variation.

Line 276-277: puzzling sentence. Did authors add the averages of regressors that vary across years or did authors replace those variables by the average values? Time invariant unobserved heterogeneity is boiling down to spatial variability or consistent differences in management practices....maybe just mention that? How does this affect rainfall and associated interactions?

Yes, the Mundlak-Chamberlain device entails modeling unobserved time invariant heterogeneity as a function of the time-averages of observed characteristics. Thus time-averages are added to the model, but not normally interpreted. This approach (which is sometimes referred to the as the Correlated Random Effects estimator) is commonly employed in applied econometric analyses, as well as increasingly in agronomic analyses (e.g. van Dijk et al. 2017, Assefa et al. 2019). Mundlak (1984), Chamberlain (1984) and Wooldridge (2010, 2019) all provide proofs that, if the core assumption is valid, the coefficient estimates converge on those of a fixed effect estimator in a large sample. We have rephrased this for clarity on page 15 of the revised manuscript.

Line 280-282: refrase this sentence please.

Rephrased for clarity on page X of the revised manuscript, which now reads as “The latter result is consistent with stylized empirical finding African smallholders are less likely to make fertilizer investments if the returns have higher levels of uncertainty.”

Line 314-316: or emphasises that technology promotion needs to be embedded in good agronomic practises. From the results, technological risks (i.e. low AEs) are dominating and translating into financial risks, but these should be dampened by good practises which is very feasible.

Rephrased on page 19 of the revised manuscript, which now reads as “This finding underscores the role that risk plays risk in smallholder decision-making, and signals that technology promotion efforts, which fail to address economic risk, are fundamentally flawed. In addition, and, conversely, agronomic practices that improve agronomic use efficiencies through technology promotion can reduce these financial risks”.

References:

References

Assefa, B.T., Chamberlin, J., Reidsma, P., Silva, J.V. and van Ittersum, M.K., 2020. Unravelling the variability and causes of smallholder maize yield gaps in Ethiopia. Food Security, 12(1), pp.83-103. https://doi.org/10.1007/s12571-019-00981-4

Chamberlain, G., 1984. Panel Data. In: In: Grilliches, Z., Intriligator, M.D. (Eds.), Handbook of Econometrics, vol. 2. North-Holland Press, Amsterdam, pp. 1248–1318.

Ichami, S.M., Shepherd, K.D., Sila, A.M., Stoorvogel, J.J., Hoffland, E., 2019. Fertilizer response and nitrogen use efficiency in African smallholder maize farms. Nutr Cycl Agroecosyst 113.

Mather, D., Minde, I., Waized, B., Ndyetabula, D. & Temu, A. (2016). The profitability of inorganic fertilizer use in smallholder maize production in Tanzania: Implications for alternative strategies to improve smallholder maize productivity (No. 1093-2016-88057). GISAIA Working Paper #4. No 245891, Food Security Collaborative Working Papers from Michigan State University, Department of Agricultural, Food, and Resource Economics. http://DOI.org/10.22004/ag.econ.245891

Mundlak, Y., 1978. On the pooling of time series and cross section data. Econometrica 46 (1), 69–85.

Rurinda, J., Zingore, S., Jibrin, J.M., Balemi, T., Masuki, K., Andersson, J.A., Pampolino, M.F., Mohammed, I., Mutegi, J., Kamara, A.Y., Vanlauwe, B., Craufurd, P.Q., 2020. Science-based decision support for formulating crop fertilizer recommendations in sub-Saharan Africa. Agric. Sys. 180, 102790.

van Dijk, M., Morley, T., Jongeneel, R., van Ittersum, M., Reidsma, P. and Ruben, R., 2017. Disentangling agronomic and economic yield gaps: An integrated framework and application. Agricultural Systems, 154, pp.90-99. https://doi.org/10.1016/j.agsy.2017.03.004

Wooldridge, J.M., 2010. Econometric analysis of cross section and panel data. MIT press.

Wooldridge, J.M., 2019. Correlated random effects models with unbalanced panels. Journal of Econometrics, 211(1), pp.137-150. 

Reviewer #2: 

This paper presents a methodological framework for addressing the question of how to target interventions to enhance the benefits of using inorganic fertilizer under high spatial heterogeneity and rainfall variability faced by smallholder farmers. The framework addresses this question by estimating the response of yields and profitability to inorganic fertilizer application using location-specific data in the case of maize in Tanzania by combining different sources of data from household surveys to spatial data. The authors use methods that I have never seen applied to issues of agronomic responses before, such as the Random Forest model. In fact, I had to find out exactly what this model entailed, so by reviewing this paper I have learned something completely new to me. In my opinion, this is a very interesting and innovative paper. It is well-written and useful. I have however a few comments.

We thank the reviewer for the positive overall assessment of our paper, and for the specific comments and suggestions, which we found very helpful. We have addressed each of these comments in detail below.

The authors mention three constraints for low fertilize usage: gap in agronomic response, low and variable economic returns, and risk averseness. However, a constraint that is not explicitly addressed is the lack of physical availability. Even if a farmer wants to buy fertilizer, nobody may sell it in the area. Obviously, this means a very high price due to transportation costs, but also search costs. So fertilizer is expensive and difficult to find and procure. In fact, one of the reasons for outside interventions is to increase the physical availability of fertilizer. So, the constraints the authors mention are demand constraints, but there they fail to mention supply constraints.

This is a good point, which we acknowledge in the revised manuscript in Line 36: “First, poor farmers’ physical access to fertilizer as a result of the heterogeneous coverage of key public goods and services”

In lines 46-48, the authors state that there has been a dearth of planning and targeting frameworks such as the one they propose in the paper. This is a very important point, but it may be useful to discuss whether there have been other types of frameworks to address the issue of targeting interventions to increase access to fertilizer.

We are not aware of other similar frameworks, although we do now acknowledge that spatial tools for targeting fertilizer recommendations do exist, e.g. the Tanzanian Soil Information System (TanSIS), in a footnote on page 3.

Also, it may be useful as part of the background to talk about programs to procure fertilizers to farmers. No mention of those explicitly. Particularly in the context of Tanzania.

We now make reference to the NAIVS program and the Government of Tanzania’s strategic objectives to raise fertilizer usage on page 4. 

Lines 13- 131. Fertilizer response was modeled with ~ 14.5 farms per district. Is this enough to capture the variability present in Tanzania?

We certainly agree with the reviewer that more data would be better. However, our sample size of 601 observations is comparable to other empirical studies (e.g. Baudron et al, 2019; Komarek et al. 2017; van Loon et al. 2019). While we believe that our sample is generally representative of the most important maize producing areas in the country, we make no formal claims of statistical representativeness. We improved the description of the sampling framework in lines 138-144.

Sine I assume that many of the readers of this paper may be like myself, i.e. not familiar with Random Forest model, I suggest, if possible, that the authors refer to a paper that provides a non-technical explanation of the model. They do cite the paper by Breiman L., but it is quite technical. I had to search the Internet to find information on the model. Providing a non-technical reference to the model will be very useful for a reader not familiar with the model.

Added reference to Liaw and Wiener 2002.

Table 1 shows that there were 601 pooled observations in the farm survey, based on 362 farms for two years. Since there where 362 households, this indicates that there were 123 missing cases (362*2= 724-601). They do not mention that and the reasons for missing data.

Added in page 7: Because of lack of measurements in the field, only 601 yield estimates from 455 households were available for modeling.

In the same table, for the variable female head of household (yes=1) it states a value of 7.067. This value does not make sense. I imagine it refers to 7%. So for consistency should be expressed as 0.07

Corrected: Years of education of head of household

Lines 152-154, could you give a short summary of the approach used. Not everybody has the time to read the original paper. I extracted this from the paper they cite:

 “We show that in many countries this variation can be predicted for unsampled locations by fitting models of prices as a function of longitude, latitude, and additional predictor variables that capture aspects of market access, demand and environmental conditions.”

We have rephrased this for clarity on page 9 of the revised manuscript.

Lines 156-160 summarizes the method to estimating farm-gate maize prices. Refers to Table A in SI text, lots of variables used. What was the number of observations of prices used?

601 observations were used. This was specified in page 9.

Lines 278-280 and Table 3 be more explicit about what (1) and (2) mean, I assume that in (1) std dev is not considered and in (2) it is. Please clarify.

Revised note under the table for clarification. 

I think that the authors could make the paper more interesting if they include in the discussion section their views on the implications of their findings for the debate on closing yield gaps. This is an important issue in the literature, and I see that their results are very relevant. They show that in fact maximizing yield is not profitable and that what is profitable will translate into maintaining a relatively large yield gap. Worth discussing.

This is a good suggestion. We have revised the manuscript, which now includes the following text on page 18: “Our analysis has direct implications for the debate on closing yield gaps. Closing yield gaps may not be economically feasible in areas which are remote from markets. However, using a framework such as the one we propose may help to identify where to prioritize investments in closing yield gaps, i.e. where returns on investment are largest.” 

In general, I think this is an very good paper that deserves to be published.

We thank you for these comments, which have helped us to improve our manuscript and make a stronger contribution.

References

Baudron, F., Zaman-Allah, M.A., Chaipa, I., Chari, N. and Chinwada, P., 2019. Understanding the factors influencing fall armyworm (Spodoptera frugiperda JE Smith) damage in African smallholder maize fields and quantifying its impact on yield. A case study in Eastern Zimbabwe. Crop Protection, 120, pp.141-150.

Komarek, A.M., Drogue, S., Chenoune, R., Hawkins, J., Msangi, S., Belhouchette, H. and Flichman, G., 2017. Agricultural household effects of fertilizer price changes for smallholder farmers in central Malawi. Agricultural Systems, 154, pp.168-178.

van Loon, M.P., Adjei-Nsiah, S., Descheemaeker, K., Akotsen-Mensah, C., van Dijk, M., Morley, T., van Ittersum, M.K. and Reidsma, P., 2019. Can yield variability be explained? Integrated assessment of maize yield gaps across smallholders in Ghana. Field Crops Research, 236, pp.132-144.

---

## [Decision Letter · Decision Letter 1]

1 Sep 2020

Fertilizer profitability for smallholder maize farmers in Tanzania: A spatially-explicit ex ante analysis

PONE-D-20-10700R1

Dear Dr. Palmas,

We’re pleased to inform you that your manuscript has been judged scientifically suitable for publication and will be formally accepted for publication once it meets all outstanding technical requirements.

Kind regards,

Luigi Cembalo, PhD

Academic Editor

PLOS ONE

Additional Editor Comments (optional):

Reviewers' comments:

Reviewer's Responses to Questions

**Comments to the Author**

1. If the authors have adequately addressed your comments raised in a previous round of review and you feel that this manuscript is now acceptable for publication, you may indicate that here to bypass the “Comments to the Author” section, enter your conflict of interest statement in the “Confidential to Editor” section, and submit your "Accept" recommendation.

Reviewer #2: All comments have been addressed

2. Is the manuscript technically sound, and do the data support the conclusions?

Reviewer #2: Yes

3. Has the statistical analysis been performed appropriately and rigorously? 

Reviewer #2: Yes

4. Have the authors made all data underlying the findings in their manuscript fully available?

Reviewer #2: Yes

5. Is the manuscript presented in an intelligible fashion and written in standard English?

Reviewer #2: Yes

6. Review Comments to the Author

Reviewer #2: I am satisfied with the way the authors addressed my comments. This is a very good and innovative paper

7. PLOS authors have the option to publish the peer review history of their article (what does this mean?). If published, this will include your full peer review and any attached files.

Reviewer #2: **Yes: **Mauricio R. Bellon

---

## [Editor Report · Acceptance letter]

11 Sep 2020

PONE-D-20-10700R1 

Fertilizer profitability for smallholder maize farmers in Tanzania: A spatially-explicit ex ante analysis 

Dear Dr. Palmas:

I'm pleased to inform you that your manuscript has been deemed suitable for publication in PLOS ONE. Congratulations! Your manuscript is now with our production department. 

Kind regards, 

on behalf of

Dr. Luigi Cembalo 

Academic Editor

PLOS ONE